# The COVID-19 Vaccination Strategy in Brazil—A Case Study

**Llanos Bernardeau-Serra** [1,*,†] , **Agathe Nguyen-Huynh** [1,*,†] , **Lara Sponagel** [1,*,†] , **Nathalia Sernizon Guimarães** [2] , **Raphael Augusto Teixeira de Aguiar** [3] and **Milena Soriano Marcolino** [4]

1   Global Studies Institute, Université de Genève, 1211 Geneva, Switzerland
2   Infectious Disease and Tropical Medicine Postgraduation Program, Medical School, Universidade Federal de Minas Gerais, Belo Horizonte 30130-100, Brazil; nasernizon@gmail.com
3   Department of Public Health, Medical School, Universidade Federal de Minas Gerais, Belo Horizonte 30130-100, Brazil; raphael@medicina.ufmg.br
4   Department of Internal Medicine, Medical School, Universidade Federal de Minas Gerais, Belo Horizonte 30130-100, Brazil; milenamarc@ufmg.br
*   Correspondence: maria.bernardeau@etu.unige.ch (L.B.-S.); agathe.nguyen@etu.unige.ch (A.N.-H.); lara.sponagel@etu.unige.ch (L.S.)
†   These authors are equal contribution to this article.

**Abstract:** Brazil is among the countries which have faced two devastating infection waves of COVID-19 in the past year. Despite the fact the country has one of the world's leading immunization programs, Brazil only slowly established a national COVID-19 vaccination strategy and campaign. This case study is based on an integrative review of primary and secondary literature sources. Different search strategies on Medline and Google Scholar were performed for the case presentation, for the management and outcome of the COVID-19 outbreak and for the state of the COVID-19 vaccination program. Official documents from the Brazilian Ministry of Health, the website of the World Health Organization and pharmaceutical companies were also reviewed. Searches were limited to English, French, German, Portuguese and Spanish. This article describes the Brazilian COVID-19 vaccination campaign and the drivers and barriers to its implementation; and evaluates further investigations needed to have a conclusive overview over the constantly evolving situation. Healthcare inequalities, which were widened during the pandemic, a lack of coordination at the federal level, the absence of federal government support for scientific research and the lack of endorsement and commitment to the mitigation of the COVID-19 pandemic set the country's COVID-19 vaccination campaign off to a challenging start. However, Brazil had a well-developed primary care system and national vaccination program prior to the pandemic, which are both important facilitators. At the time of writing, six vaccines are currently available in the country, and the program is advancing. The scientific community needs to continue to investigate the country's vaccination strategy and its implementation to make sure that maximum effort is undertaken for the health of the Brazilian population.

**Keywords:** Brazil; COVID-19; disease management; mass vaccination; pandemic

## 1. Introduction

With the emergence of the novel SARS-Cov-2-virus (causing COVID-19) and its rapid spread across the world, the race for a safe and efficient vaccine began. Brazil was the first country in Latin America to report a case of COVID-19 on 26 February 2020 [1]. With a naive population (with no immunity), lack of effective pharmaceutical interventions and the inconsequential application of non-pharmaceutical interventions, the number of cases and deaths in Brazil increased rapidly [2].

Even though there were attempts by the authorities to contain the spread of the virus with non-pharmaceutical interventions (such as physical-distancing measures, mask wearing, hand-sanitation, quarantines or travel restrictions), the unhalted spread of the virus (R0 = 3) led to overcrowding in hospitals, which overwhelmed Brazil's healthcare

systems [3]. This highlighted the fact that the most effective epidemiological intervention remained the development of a safe and effective vaccine, which could be produced fast and distributed quickly to a large part of the population [4,5]. As soon as the genetic sequencing of the SARS-Cov-2-virus was published on 11 January 2020, different actors started engaging in a large effort towards the research and development of a COVID-19 vaccine [6].

On 16 March 2020, the first vaccine entered clinical trials (Moderna's mRNA 1273) [7]. In the spring/summer of 2020 China (June) and Russia (August) approved their domestic vaccines CanSino and SputnikV, respectively, for emergency use in their populations [8,9]. On 31 December 2020, the WHO issued its first emergency use approval for a COVID-19 vaccine (Pfizer/BioNTech's Comirnaty) [10]. At the time of writing, there are 17 vaccines currently in use in the world, 108 in Phase I, II, or III of clinical trials and another 184 in preclinical stages [11,12]. Moreover, as of 18 July 2021, according to the WHO, over 3.6 billion doses of COVID-19 vaccines have been administered, in every region and in almost every country of the world. However, there are large discrepancies between countries in the vaccine rollout [13].

The aim of this study is to describe the development of the COVID-19 vaccination strategy and the current state of the vaccination progress in Brazil. We will provide a presentation of the management and outcome of the COVID-19 outbreak in Brazil, i.e., the epidemiological situation during the pandemic, the non-pharmaceutical interventions, the impact on the economy, the social and political disruptions, as well as an in-depth description of the state of the COVID-19 vaccination in Brazil, i.e., the characteristics of the available vaccines, the registration procedure for the vaccines, the production and purchase schedule, logistics (storage, delivery), the prioritization of target groups and clinical management. The overall management of the vaccination strategy and the current state (15 July 2021) of the vaccine administration and distribution will then be discussed.

## 2. Materials and Methods

This case study is based on a integrative review of scientific literature, performed using Medline and Google Scholar and the following keywords: for the case presentation "Brazil, geography, demography, ethnicity, ethnic groups, census, demographic distribution, government, social inequalities, regional inequalities, geographical disparities, politics, democracy, president, healthcare system, universal health system, Unified Health System"; for the management and outcome of the COVID-19 outbreak "Brazil, COVID-19, cases, deaths, underreporting, management, Manaus, immunity, non-pharmaceutical interventions, social distancing, physical distancing, government measures, state measures, economy, economic growth social disruption, political disruption, politics, Bolsonaro". Lastly, for the state of the COVID-19 vaccination program, we used the keywords "vaccine, vaccination strategy, immunization program, management, vaccine, approval, distribution, coverage, prioritization, target groups", we performed an in-depth review of official documents from the Brazilian Ministry of Health (vaccination plans, press releases) and we explored the website of the WHO and pharmaceutical companies.

All the data are primary and secondary literature sources and are focused on the case of Brazil. As it is a descriptive, non-analytic, single case study there is no hypothesis testing. The literature included is in English, French, German, Portuguese and Spanish. This limits the scope in terms of available and included papers but as it is a case study in Brazil and literature in Portuguese was considered, this appears to be a reasonable limit of this study.

The development and distribution of the COVID-19 vaccination are very dynamic processes and the global situation changes at a fast pace. This case study is a snapshot of the vaccine-development situation up to July 2021.

**3. Case Study**

*3.1. Case Presentation*

3.1.1. Demographic, Geographic, Economic, and Social Characteristics of Brazil

The Federative Republic of Brazil is the fifth largest country in the world, and the largest and most populated country in South America, with an estimated population of 212.6 million in 2020 (2.7% of the world's population), unequally distributed in an area of 8.4 million km$^2$, divided in five main geographical areas [14–17].

Economic power is also unequally distributed among those regions. The south and southeast regions are the hub of the industrial and economic activities, accounting for around 20% and 60% of the national Gross Domestic Product (GDP), respectively [18,19]. On the other hand, although the Northeast accounts for 28% of the population and one-fifth of the national agricultural production, the development of the industrial and services sectors is far from the southern and southeastern levels [19]. Although the population is divided in five groups in the self-reported census (white representing 46%, *pardo* 44%, black 9%, Asian 1% and indigenous 0.4%), the population is highly miscegenous [20,21].

Although Brazil is an upper middle-income country, among the strongest emerging market economies of the world [18,22,23], it is also amongst the top ten most unequal countries, and these inequalities are geographically defined [24]. The human development index (HDI) was at 0.765 in 2019, but it falls to 0.570 when inequalities are accounted for [14]. The northern regions of the country are also the ones with the lower HDI, and the southern regions the ones with the higher HDI [25].

Twenty percent of the population lives below the poverty line for the Upper Middle-Income Class, set by the World Bank at USD 5.50 (2011 Purchasing Power Parity) per day per capita and 4.4 % of the population is below the international poverty line of USD 1.90 per day per capita [26]. Around 12 million people live in so-called "favelas", overcrowded slums with no proper water and sanitation services [27].

3.1.2. Political Characteristics of Brazil

Brazil is a federal presidential representative republic (constitutional republic). The country has a multiparty election system, which is composed by the government, the 26 states, the federal district and the municipalities. The president is in charge of the national administration and unifies the executive power of the nation in one position by being the head of state and the head of the government at the same time [28]. He or she is free to form his or her cabinet initially and change it throughout the course of the presidency. It is also possible for the president to create or abolish ministries. Therefore, the Brazilian president has an ambivalent power-position, which is unusual in democracies [29].

Since 2018, Jair Messias Bolsonaro has been the president of Brazil. Already before the presidential elections in 2018, Bolsonaro was a contested political figure and his controversial policies were subject to many different interpretations and narratives [30]. Prior to his election, Brazil was shaken by a national corruption scandal. Concurrently, the country fell into an economic crisis, which was fueled by investment-deficits in the health- and education sectors of previous administrations. Unemployment rose to a historic high, poverty and crime rates started to grow, and the population increasingly lost their trust in the political administration [31]. Bolsonaro made use of the political vacuum to position himself as a political outsider, detached from the corrupt system [30].

3.1.3. Brazil's Healthcare System

The Brazilian healthcare system has three subsectors: the public subsector, financed with taxes and social contributions; the private (non-profit and for-profit) subsector, financed with both public and private funds; and the private health insurance subsector, composed of different forms of health plans, health insurance premiums and tax subsidies. These three sectors are linked, and people can access them according to their purchase capacity [32].

The public health system, *Sistema Único de Saúde* (SUS) is one of the largest health systems in the world [33]. The management and provision of healthcare is decentralized. Thus, decision-making and funding power is not only given to the federal government, but also to the states and municipalities [34]. The SUS covers every level of healthcare, including primary and emergency healthcare, and follows the principles of universality, integrality and equity [35]. That way, it is meant to ensure complete, universal and free access to healthcare for the whole Brazilian population. It also promotes vaccination, and both epidemiological and sanitary surveillance [36]. In fact, it has one of the best vaccination programs of the world: the Brazilian National Immunization Program (*Programa Nacional de Imunizações*, PNI) [37,38]. The PNI is coordinated by the Ministry of Health (MoH) and it covers the whole Brazilian population. Brazil is one of the countries with the most free of cost vaccines in the world [39].

Nowadays, the SUS is still struggling to ensure universal coverage, as its implementation has faced political and economic constraints "which promoted a neoliberal rather than a universal approach" [40,41]. Preexisting socioeconomic disparities are being reinforced through unequal access to healthcare services [42].

The groups with high social vulnerability can be divided into: Indigenous People, Traditional African Communities and Traditional *Ribeirinhas* Communities. The Ministry of Social Development has a special Committee for the Traditional People and Communities, which include the Indigenous People, the Traditional *Quilombolas* Communities (the Traditional African Communities, descendants of African American slaves), and the Traditional *Ribeirinhas* Communities (which means settled along rivers), among others [43,44]. "Indigenous Peoples living in Indigenous Lands" have higher risk of infectious comorbidities, higher burden of diseases linked with environmental contamination and higher incidence of chronic diseases. These Indigenous communities also have a higher burden of infectious diseases due to "their collective way of living", to the "difficulties faced regarding the implementation of non-pharmacological measures" and to their geographic location, having to travel long distances in order to access a healthcare service [38]. The *Quilombolas* and *Ribeirinhas* populations also suffer from a higher risk of infectious transmission due to high levels of cohabitation [38].

### 3.2. Management and Outcome of the COVID-19 Outbreak

3.2.1. Epidemiological Situation of the Country Regarding COVID-19

Brazil has the second highest death toll from COVID-19 after the USA and the third highest cumulative number of detected cases after the USA and India, i.e., almost 20 million reported cases of COVID-19 and over 0.5 million deaths from COVID-19 as of 17 July 2021 [45].

The first case of COVID-19 was reported in the city of São Paulo, on 26 February 2020. Cases and deaths increased steadily until a first peak was reached in July 2020 with over 45,000 confirmed cases and 1000 deaths per day [2]. The daily number of cases and deaths then decreased until November 2020 when they dropped below 10,000 and 400, respectively. After this drop, the 'second wave' of the pandemic started, particularly from January 2021 when Manaus (capital of state of Amazonas) was hit by a wave of COVID-19 cases. The second wave received international attention, as it occurred despite the population of Manaus having been estimated to exceed the herd immunity threshold of 67% [46]. This resurgence has been hypothesized to be at least partly due to the presence of the variant of concern 202101/02 or lineage P1 [47]. This variant was first detected in Manaus but has rapidly spread around the world, mostly in Latin America [47] and it seems to be more transmissible than the original COVID-19 strain circulating [48]. From January onwards, cases and deaths increased exponentially. Brazil seems to have reached the peak of its long and violent second wave at the end of March/beginning of April 2021, with a maximum of over 77,000 daily new cases (on 27 March 2021) and of over 4,000 daily deaths (12 April 2021). Subsequently, the number of cases started to decrease, but rose again to over 77,000 daily new cases and over 2000 daily deaths in June 2021. At that time, only 12%

of Brazilian population was fully vaccinated. From then, the number of cases progressively decreased to less than 40,000 daily new cases on 17 July 2021 (Figures S1 and S2) [2,49].

It is important to note that the aforementioned reported cases and deaths from COVID-19 are underestimations, due to a lack of testing [50,51]. Indeed, testing in Brazil is largely below the world average [52]. The main reason for this was an initial shortage of tests and reagents due both to the increase in global demand and the lack of coordination of reagent purchases by the country, which forced the federal government to prioritize testing in the public subsector for symptomatic patients who were hospitalized only. Brazil was eventually able to circumvent this issue, but it still faces a dependency on the importation of diagnostic components, as well as inequalities between the public and the private sector regarding the funding and distribution of tests [53].

Results from EPICOVID-19, the largest epidemiological study of COVID-19 in Brazil, reported a six-fold difference between their estimates on the real number of infected people and official statistics [54]. Additionally, the epidemiological situation of COVID-19 in Brazil has not affected the whole population in an equal way. People in the poorest socio-economic quintile, people living in crowded conditions and Indigenous People had a much higher prevalence of and mortality from COVID-19 [55,56].

Figure S3 shows the COVID-19 epidemiological situation in Brazil as of July 2021.

### 3.2.2. Non-Pharmaceutical Interventions Undertaken by Health Authorities

In the first few months of the COVID-19 pandemic, in the absence of approved pharmaceutical interventions, non-pharmaceutical interventions (NPIs) such as physical distancing and mask-wearing were the only possible measures governments could implement [57]. When the first case of COVID-19 was detected in Brazil at the end of February 2020, the federal government's initial reaction was to quarantine only the detected cases. No other containment measures, such as contact tracing, were imposed at the national level and neither the MoH, nor any other federal government agency had developed a national strategy for the management of COVID-19 [58]. Moreover, in the public system, testing was restricted to only the severe cases of COVID-19 [59]. A SIQR (Susceptible, Infectious, Quarantine, Recovered) modelling study using government's data on COVID-19 incidence in the first two months of the COVID-19 pandemic in Brazil estimated that for every quarantined COVID-19 case, another ten infectious individuals were present in the population. As mentioned above, this resulted in an exponential increase of COVID-19 cases with all states reporting cases of the disease a month after the first detection [60].

Despite the lack of federal coordination, states started implementing physical distancing measures in the latter half of March, 2020. State authorities implemented these measures early, for the majority having detected less than 10 cases and prior to reporting the first death from COVID-19 on their soil [61]. Suspension of events, quarantine of high-risk groups and school closures were implemented in all states between 11–23 March 2021. Most states also implemented travel restrictions between states and/or municipalities. As of July 2020, seven states had recommended to their whole population to quarantine. This remained a recommendation rather than an obligation. Nineteen of the 26 states had implemented a partial economic lockdown and seven a full economic lockdown, including the suspension of all non-essential economic activities. Interestingly, five out of the seven states which implemented a full economic lockdown were in the north and northeast regions, and these tended to implement physical distancing measures earlier in the epidemiological situation than southern regions, i.e., when they did not have any reported cases or had fewer [61]. Measures varied significantly between states. However, according to the Oxford COVID-19 Government Response Tracker (OxCGRT), the stringency of the overall country's measures was estimated to be above 70% from end of March until end of August and over 60% from September 2020 to April 2021 (with a drop to the over 50% category in November) [62]. These measures seemed to be effective in moderating the exponential growth of cases according to mathematical modeling studies in Rio de Janeiro and São Paulo [3,63].

It is important to note that despite the lack of coordination at the federal level in the first months of the pandemic, Brazil has strong democratic institutions, which have supported an evidence-based management of the sanitary crisis [55]. On 15 April 2021, the Supreme Court officially delegated the management of the spread of COVID-19 by assigning the authority to implement physical distancing measures to states, districts and municipalities [64]. The SUS, despite its aforementioned flaws, has been continuously working to ensure healthcare for COVID-19 patients during the pandemic, with similar mortality rates in public and private hospitals when controlling for patient characteristics, delayed access to healthcare, and the criteria for hospitalization [65,66]. This was mostly coordinated at the state level, which has made significant efforts to increase bed capacity in ICUs and to effectively allocate cases [58].

### 3.2.3. Expected or Observed Impact on the Country Economy

The outbreak of COVID-19 in Brazil has significantly and negatively impacted its economy. The GDP Annual Growth Rate was between 1.2% and 1.6% in 2019, the year prior to the pandemic. In 2020 it decreased strongly to −0.3% in the first quarter, −10.9% in the second quarter, −3.9% in the third quarter and −1.1% in the fourth quarter. It is expected to increase from −1.1% in the first quarter of 2021, to 5% in the second quarter and then again decrease in the third and fourth quarter with 3.5% and 3.2%. In 2022 the GDP Annual Growth Rate is expected to stabilize around 2.5% [67]. Additionally, the global pandemic had strong impacts on international trade, which led to a fall in commodity prices, currency devaluations and disruptions in the supply chain (this is especially crucial for the production and import of vaccines, medical equipment and drugs) [54]. During the first quarter of 2021, the Brazilian real lost 10% of its value per US dollar compared to the beginning of the new year and came close to the 5.88 per US dollar (which was the lowest drop in 2020, the first year of the COVID-19 pandemic) [68].

The levels of national unemployment have been steadily increasing since January 2020 (11.2% national unemployment) until they reached a peak in September 2020 (14.6%), and then it declined slightly until December 2020 (13.9%) but has been increasing since the beginning of 2021 (14.4% in February 2021). The northern regions were more heavily impacted [69]. Moreover, many people work in the informal sector with no social protection [70].

Poor housing conditions, high population density and lack of clean water and proper sanitation increase the risk of infection. These are all reinforced by the missing universal coverage of the social institutions of the Brazilian state [54]. The COVID-19 pandemic has reinforced those pre-existing inequalities in income, socio-economic status and infection–prevention privileges [71].

### 3.2.4. Social and Political Disruption of COVID-19

Brazil has been significantly disrupted by the lack of coordination and disagreements between states and the federal authority. As mentioned, the states implemented NPIs quite rapidly while President Jair Bolsonaro repeatedly spoke out against these measures with the justification that the economy should be prioritized, saying for example that unemployment was worse than COVID-19, and by instead promoting early treatment, which was not scientifically proven [58,72,73]. These arguments understandably received support from a large part of the Brazilian population, a country where many people are in a precarious economic situation [58]. Disagreements on the management of the pandemic also led to an unprecedented turnover in the government. The country had three ministers of health in four weeks, two of them fired due to disagreements with the president on policies regarding physical distancing and the use of hydroxychloroquine as a treatment for COVID-19 [58]. Bolsonaro consistently minimized the potential health consequences of COVID-19, calling it a "a little flu" and saying "only the elderly are at risk" [74]. In terms of mask-wearing, Bolsonaro did not support or even discouraged it in his public speeches and meetings [75]. There was no effort by the federal government to negotiate with the

industry the acquisition of masks at a reasonable price, leading to an increase in price per mask from BRL 4.50 (USD 0.83) in January to BRL 140 (USD 25.71 by March 2020 [74]. Bolsonaro's administration also did not support the Brazilian scientific community, by for example withdrawing the budget for the largest epidemiological study on COVID-19 in Brazil after it revealed evidence of underreporting in official statistics and highlighted the socio-economic and racial inequalities in COVID-19 prevalence [76].

The federal authorities also promoted pharmaceutical interventions which had little to no scientific basis and even proved to be potentially harmful to COVID-19 patients due to severe adverse effects [58,77,78], such as the antimalarial drugs chloroquine, hydroxy-chloroquine and ivermectin. Indeed, there has been evidence of an increase in all-cause mortality in COVID-19 patients who received chloroquine/ hydroxychloroquine in randomized clinical trials [74]. Its use for COVID-19 patients has led to a shortage for malaria patients and patients with chronic inflammatory disorders for whom they are effective [79]. With regard to ivermectin, there was an increase in patients with hepatic toxicity, in some cases with the need of emergency liver transplantation [80]. During the violent outbreak of COVID-19 in Manaus, rather than advising a full lockdown of the city, the government sent chloroquine and other drugs such as ivermectin [76].

With the implementation of the vaccination strategy, the country faced an additional challenge, as the government did not ensure the supply of sufficient syringes and needles to carry out a nationwide vaccination campaign in the beginning [76].

Vaccine acceptance in Brazil is generally higher than in other countries because of the country's renowned immunization programs and campaigns before the pandemic. Despite the fact that president Jair Bolsonaro discredited the vaccination campaign, which had an impact on vaccination acceptance, a recent study has shown that only 10% of the Brazilian population showed reluctance to receive a COVID vaccine versus up to 40% in other countries [81]. People at lesser risk of severe cases of COVID-19 and supporters of political leaders who have minimized the effect of the pandemic, such as Donald Trump or Jair Bolsonaro, were less inclined to support vaccination. Another factor in vaccine hesitancy identified was the provenance of the vaccine [81]. The same study has shown that vaccines produced in China or Russia were less likely to be accepted by the Brazilian population: the acceptance decreased by 21.3% for the "Chinese vaccine" and by 15.7% for the "Russian vaccine", even though the "Chinese vaccine" is produced in Butantan Institute, Brazil. This was fueled by Bolsonaro, who stated in October 2020 that Brazil would not purchase vaccines from China because it was "not safe because of its origin" and cancelled the order of 46 million CoronaVac Vaccine, which was later restored [81]. He also stated that he would refuse to be vaccinated with the United States Pfizer vaccine, as the contract mentioned the producer would not be held responsible for any adverse effects. He dealt with the situation ironically, mentioning that there was a possibility of people turning into crocodiles after receiving the vaccine [82].

However, since the beginning of March 2021, Bolsonaro's approach to and engagement with the vaccination campaign has changed. He has started to endorse the vaccination campaign and signed measures allowing the production of a COVID-19-vaccine in Brazil and the speedup of the purchase of vaccines. This change of opinion came shortly after the acquittal of the former president Luiz Inácio Lula da Silva, the principal opponent of Bolsonaro for the 2022 election, who has openly criticized Bolsonaro's handling of the COVID-19 crisis and called on the population to be vaccinated [83].

On April 27 2021, the Federal Senate of Brazil installed a parliamentary commission of inquiry, the "COVID-19 CPI", to investigate the federal government's actions and omissions in the management of the COVID-19 pandemic [84]. The investigation is still ongoing, but it has already found that vaccine procurement and purchasing was plagued with corruption.

*3.3. Vaccination Strategy*

3.3.1. Characteristics of Available Vaccines

As of 18 July 2021, Brazil has approved six vaccines (Table 1). The first two vaccines to be approved for emergency use on 17 January 2020 were Coronavac and AstraZeneca.

Coronavac, developed by the Chinese company Sinovac, is an inactivated vaccine. Similarly to the vaccines for polio, rabies or hepatitis A, it is created from inactivated coronaviruses, which can no longer replicate but whose spikes remain intact, allowing the body to create an immune response [85]. It requires two doses and needs to be kept in a standard refrigerator (2–8 °C) [86]. In July 2020, Phase III clinical trials of this vaccine were carried out in Brazil [87] and a press release from Sinovac revealed a 50.65% efficacy against infection with or without symptoms [87]. However, another Phase III trial in Turkey found it had a higher efficacy of 83.5% against infections with symptoms [88]. At the time of writing this paper, Sinovac has published Phase I and II clinical trials in *The Lancet* [89] but have yet to publish their Phase III clinical trials in peer-reviewed journals [90]. It has not yet received standard registration in Brazil but is in the process of doing so [91].

Vaxzevria, which is also known as AstraZeneca, AZD1222, or Covishield when it is produced in India, was developed by the University of Oxford and the British–Swedish company AstraZeneca. It received standard registration from Brazil on March 12th, 2021 [92]. It is a viral vector vaccine, relying on a more recent technology. Rather than introducing antigens in the body, Vaxzevria delivers a modified harmless virus that contains the mRNA sequence of the antigen of the SARS-Cov-2 virus. This harmless virus acts as a delivery system conducting human cells to produce spike proteins, which in turn trigger an immune response [92]. Vaxzevria requires two doses and needs to be kept in a refrigerator. Phase III clinical trials in the USA have revealed a 76% efficacy against infections with symptoms [93]. Phase III clinical trials have also been carried in Brazil, South Africa and the UK and found efficacy between 55·1% and 81·3% according to the time interval between the two doses, <6 weeks and ≥12 weeks, respectively [94,95]. This vaccine is sold at low prices and Oxford/AstraZeneca have pledged that they will sell it at cost price, i.e., on a non-profit basis, to low- and middle-income countries [96]. Moreover, AstraZeneca was the first pharmaceutical company to join the COVAX facility, the vaccines pillar of the Access to COVID-19 Tools (ACT) Accelerator, which aims to accelerate the development, production, and equitable access to COVID-19 tests, treatments, and vaccines [97]. In partnership with the Serum Institute of India (SII) producing the Indian version of the vaccine called Covishield, this vaccine represents, as of April 2021, the largest contributor to COVAX, i.e., 37 million doses out of 38 million distributed [98]. However, there has been controversy about the side effects of the vaccine, specifically on blood clots in young people, leading some European countries to suspend its use, even though it seems to be a very rare complication [99].

On 3 February 2021 the vaccine Comirnaty (also known as tozinameran or BNT162b2) developed by the American company Pfizer and the German company BioNtech received full authorization in Brazil [91]. This vaccine is an mRNA or messenger RNA vaccine, which uses the same technology as a viral vector but directly inserts the genetic code into the cell rather than using a virus [100]. Phase II and III clinical trials were carried out in Brazil, among other countries [101]. The efficacy of the vaccine is the highest so far, as clinical trials results have shown a 91.3% vaccine efficacy and it has been shown to be effective in South Africa, where the B.1.351 lineage is prevalent [102]. The vaccine requires two doses, three weeks apart and needs to be kept in a freezer at –13 °F to 5 °F (–25 °C to –15 °C) [103]. Pfizer and BioNTech have reached an agreement with COVAX to provide up to 40 million doses in 2021 [104].

On 31 March 2021 Brazil approved for emergency use the viral vector vaccine Ad26.COV2.S, also known as Janssen, developed by the American company Johnson & Johnson. Phase III clinical trials have been carried in Brazil among other countries [91]. The vaccine has been shown to have 66.9% efficacy two weeks after administration [95]. This vaccine requires only one dose. It can be kept for two years when frozen at −4 °F (−20 °C), and three months when

refrigerated at 36–46 °F (2–8 °C), which means it can more easily be shipped and stored than mRNA vaccines [105]. It is, similarly to the AstraZeneca vaccine, available on a not-for-profit basis for emergency pandemic use [106] and COVAX has secured 500 million doses through 2022 [107].

In June 2021, Anvisa approved, under special conditions, limited importation of two COVID-19 vaccines: Covaxin [108,109] and Sputnik V. Covaxin is an immunizer BBV152, which is a whole-virion inactivated SARS-CoV-2 vaccine formulated with a Toll-like receptor 7/8 agonist molecule adsorbed to alum (Algel-IMDG) or alum (Algel). The Indian company Bharat Biotech, in collaboration with the National Institute of Virology and the Indian Council of Medical Research, produced this vaccine. It was approved for clinical trials for clinical testing in June 2020, with phase I and II results released in December last year [108,109].

Sputnik V is a recombinant heterologous adenovirus (rAd), Gam-COVID-Vac manufactured by União Química—a Brazilian company partnered by the Russian Direct Investment Fund (RDIF) [110]. Denis Logunicov and colleagues report their interim results from a phase 3 trial of the Sputnik V COVID-19 vaccine in *The Lancet*. The trial results show a strong and consistent protective effect in all participating age groups [111].

At the time of writing, Brazil is in the process of receiving authorization to test its own domestically developed vaccine. Indeed, Butantan Biological Institute has requested the regulatory agency of Brazil to start human clinical trials for their vaccine ButanVac [112]. Another 17 potential vaccines are currently under development in pre-clinical stages in Brazil [91]. These are often financed by municipalities due to a lack of funding from the federal government [113]. For example, clinical trials of the potential Spintec vaccine, which is being developed by researchers at the Federal University of Minas Gerais to combat COVID-19, has received USD 5.9 million [114].

### 3.3.2. Registration of Vaccines Procedure

The vaccines had to go through a regulatory process with the *Agência Nacional de Vigilância Sanitária* (ANVISA), the regulatory agency of the government of Brazil [115]. An authorization of emergency use for an experimental vaccine is a mechanism used by ANVISA that facilitates the availability and use of vaccines during a specific period of time and to a targeted population due to emergency reasons. This tool can be put in place even if the vaccines have not been evaluated for registration in the country, as long as they meet the requirements for safety, quality and efficacy [116]. Through the *Resolução da Diretoria Colegiada* (RDC) 465/2021, ANVISA also regulated the exemption of registration and authorization for emergency use for the vaccines acquired by the MoH under the COVAX Facility [117].

On 10 December 2020 the MoH and ANVISA released an official resolution establishing the temporal authorization of COVID-19 vaccines for emergency use: the RCD 444/2020 [118]. This resolution legislates the submission of temporary authorization of a vaccine for emergency use [118]. In order to facilitate this process, ANVISA released the Guide nº 42/2020 on "minimum requirements for submitting an application for temporary authorization for the emergency use of COVID-19 vaccines on an experimental basis", a guide that specifies the required procedures and documents that a company needs to provide in order for their submission to be taken into account [119]. This compilation of documents is called the Clinical Drug Development Dossier, and it is submitted to ANVISA in order to support the registration and authorization for emergency and temporary use of vaccines against COVID-19 in their experimental phase [116]. Once the information is provided, ANVISA reviews the data submitted to check that the vaccine meets the required criteria.

As previously specified, in Brazil, the health registration of the product is the only instrument to use the vaccine in the general population without restricted time or target population, and the allowance for emergency use cannot be considered as a substitute for the latest registration [119]. Even though the registration process for a vaccine is

mandatory in any case, when the technological development of a vaccine is conducted entirely abroad, there is no need for ANVISA's prior consent for conducting clinical trials in the country [120].

To conduct a standard registration, the first step is the manifestation of interest that needs to be submitted to ANVISA by the company producing the vaccine. The essential documentation required to conduct this process and in order to guarantee the safety, quality and efficacy of a biological product, is specified in the RDC 55/2010 [121]. Among others, the company must submit the Good Manufacturing Practice Certification (GMP), the Operating Authorization (AFE), the justification for registration, a pharmacovigilance plan, and a report with data on the raw materials used in the vaccine. The request for registration can take place while the company is conducting phase III clinical trials, therefore having already conducted phase I and II with success. In addition, the company must conduct a drug stability study to generate data on the shelf life and appropriate storage conditions and must ensure the monitoring of new adverse reactions [115]. The prioritization of the analyses of applications for registration and authorization for clinical trials by ANVISA is regulated under the RDC 204/2017 [122]. Considering the current pandemic, ANVISA released the RDC 348/2020 [123] and its update, RDC 415/2020 [107], which reinforced the prioritization of COVID-19 related products. The status of each vaccine's documentation, for both registration and authorization for emergency use, is published by ANVISA and can be checked on their website [124]. In order to speed up the process of registration of vaccines, ANVISA created the "continuous submission", approved in the Normative Instruction 77/2020 [125]. This continuous submission allows companies responsible for the vaccines to submit the required data and technical documents for the vaccine registration while the data and documents are generated, creating a continuous exchange of information between the company and ANVISA.

On 17 January 2021, ANVISA authorized the COVID-19 vaccines of Sinovac/Butantan, and AstraZeneca/Fiocruz for emergency use. On 23 February 2021, ANVISA authorized the registration of the Pfizer/Wyeth vaccine [126], and on 12 March 2021, the AstraZeneca/Fiocruz vaccine. In June 2021, ANVISA authorized the COVID-19 vaccines of Sputnik [127] and Covaxin [128] for emergency use. The National Vaccination Campaign against COVID-19 started on 18 January 2021 [36].

### 3.3.3. Production/Purchase Schedule for Vaccines, Vaccine Supply Volume and Its Dynamics

Brazil has been receiving vaccine doses from different suppliers, and is also producing and planning to produce doses on its soil. The first two vaccines which were used in Brazil came from two separate agreements made in June 2020: one agreement between the state-run Butantan biomedical Institute in São Paulo and Sinovac and another agreement between the federally funded Oswaldo Cruz Foundation (Fiocruz) and AstraZeneca [129]. Both agreements also included a transfer of technology with Butantan and Fiocruz having increasing responsibility in the production process of the doses [129].

Shortly after Sinovac and AstraZeneca's vaccines were approved for emergency use in Brazil by ANVISA, on 17 January 2021, the first dose of Sinovac was administered [91]. Indeed, Butantan had already started to receive Sinovac's Coronavac Active Pharmaceutical Ingredients (APIs) and to take care of the bottling, labelling and quality control [130]. As of the time of writing, Fiocruz provided the majority of doses used in the PNI [131]. In January, they delivered 8.7 million doses of the vaccine; in February, 4.8 million; in March, 22.7 million; in April, 5.8 million doses; and in July, 1 million doses [132]. Five days after the approval, on 22 January 2021, 2 million doses of ready-to-use Covishield, the AstraZeneca vaccine produced by the Serum Institute of India, arrived by plane in Brazil [133]. As of the 15 March 2021, Brazil had received 4 million doses in total of Covishield [134]. On 8 March 2021, Fiocruz announced the start of a large-scale production, i.e., with the goal of producing one million doses per day in their production facility called the Immunological Technology Institute [135]. On 12 March 2021, they received approval from the ANVISA to do so. According to the Foundation, they would receive enough APIs from China (four

batches of 256 L each) to produce approximately 30 million doses of vaccine, guaranteeing their production until the end of May [134]. A new shipment of 6000 liters of APIs arrived in Brazil in June 2021, allowing for the production of 10 million more doses, Butantan said [136].

Brazil is also receiving doses through the COVAX Facility, a collaboration co-led by the WHO, Gavi and the Coalition for Epidemic Preparedness and Innovations (Cepi). The collaboration aims to accelerate the development, production, and equitable access to COVID-19 tests, treatments, and vaccines [97]. It aims to provide 2 billion doses by the end of 2021, allowing countries to vaccinate between 10-20% of their population. Brazil's economic situation does not make it eligible for GAVI COVAX Advance Market Commitment (COVAX AMC) which provides low- and middle-income economies with donor-funded COVID-19 vaccines, regardless of ability to pay. In the COVAX scheme, Brazil is a self-financing country, i.e., they invested through the COVAX facility for Research and Development of COVID-19 vaccines and will therefore receive doses according to their investment [137]. On the 24 September 2020 President Jair Bolsonaro authorized the federal executive branch to adhere to COVAX under the Optional Purchase Arrangement, in which participants can choose to opt out of receiving any vaccines [138]. Brazil has earmarked USD 454 million for securing vaccines through COVAX and is therefore entitled to 9,122,400 doses, which should cover around 10% of the population [139]. On 21 March 2021, Brazil received 1,022,400 doses of the AstraZeneca/Oxford vaccine—manufactured by SK Bioscience (South Korea)—through the COVAX mechanism [140]. This first delivery was the largest vaccine acquisition and supply operation in global history [141].

Additionally, on the 19 March 2021, Brazil has announced an agreement with Pfizer/BioNTech and Johnson and Johnson to acquire 100 million doses of Comirnaty BNT162b2 and 38 million doses of Janssen [142]. Comirnaty is an mRNA vaccine, which was approved by ANVISA for standard registration on the 23 February 2021 [91]. In April 2021, Brazil received the first 1 million doses of Comirnaty. Jannsen is a viral vector vaccine, which was approved for emergency use by ANVISA on the 31 March 2021. Lastly, Brazil has signed agreements with Precisa/Bharat Biotech (Covaxin vaccine) and União Química/Gamaleya (Sputnik V vaccine) [142]. Both vaccines had received authorization for exceptional import by ANVISA on 5 June 2021 under some restrictions: only 4,000,000 Covaxin doses (out of previously requested 20,000,000) could be imported. A smaller number of Sputnik doses could be sold to Brazil under that decision: only 670,192 doses, which could be used only in six states. As of 19 July 2021, there were no official data registering delivery of any of them [143]

Table 1 provides a synthesis of the different vaccines, their characteristics (type, efficacy), their approval status, the date on which they were approved, their production sites (when known), the number of doses acquired at the time of writing in Brazil and the number of doses planned to be acquired.

**Table 1.** List of vaccines available or planned to be available in Brazil as of the 18th of July 2021.

| Name (Company's Name) | Type | Efficacy Rate in Phase III Trials (95% Confidence Interval) | Approval Status by the Brazilian Health Regulatory Agency (ANVISA) [86] | Final Production Site(s) (Bottling, Labelling and Quality Testing) | Number of Doses Acquired | Number of Doses Agreed On |
|---|---|---|---|---|---|---|
| Vaxzevria (Oxford–AstraZeneca) | Viral Vector | 76% [CI, 68.0, 82.0] in USA [93] between 55.1% (CI, 33.0, 69.9) and 81.3% (CI, 60.3, 91.2) according to the time interval between the two doses, <6 weeks and ≥12 weeks, respectively [94,95] | Emergency use (17/01/2021) Standard registration (12/03/2021) | Serum Institute of India (India) called Covishield | 4 million [134] | 12 million agreed (current delays due to the epidemiological situation in India) [134,144] |
| | | | | SK Bioscience (South Korea) | 1 million doses acquired through COVAX [141] | 9'122'400 doses allocated to Brazil through COVAX [139] |
| | | | | Fundación Oswaldo Cruz (Fiocruz) | 69 million [145] | 100 million doses (by July 2021) [13] |
| Comirnaty BNT162b2 (Pfizer/BioNTech) | mRNA | 91.3% (CI, 89.0, 93.2) [104] | Standard registration (23/02/2021) | Baxter International in Halle (Germany) [146] | 16 million [145] | 100 million agreed (between July and September) [91] |
| Coronavac (Sinovac) | Whole-virus/ inactivated virus | 50.65% in Brazil [87] 83.5% in Turkey [88] | Emergency use (17/01/2021) | Butantan | 54 million [145] | Not found |
| Janssen (Johnson & Johnson) | Viral Vector | 66.9% (CI, 59.0, 73.4) two weeks after administration [147] | Emergency use (31/03/2021) | Unknown at the time of writing [148] | 4.5 million [145] | 38 million agreed (between August and November 2021) [148] |
| Covaxin BBV152 (Bharat Biotech) | Inactivated virus | 77.8% (95% CI; 65.2–86.4) [143] | Emergency use (04/07/2021) | Bharat Biotech [108] | Ongoing | Not found |
| Sputnik V | heterologous recombinant adenovirus (rAd) [110] | 91.6% (95% CI 85.6–95.2) [110] | Emergency use (04/07/2021) [127] | União Química—a Brazilian company partnered with the Russian Direct Investment Fund (RDIF) | Ongoing | Not found |

### 3.3.4. Organization of Storage and Delivery of Vaccines

According to the national legislation, the management of the vaccination campaign is shared between the three spheres of the Brazilian Health System: the federal level, the state level and the municipal level. The Brazilian MoH is responsible for the coordination between these three spheres as well as for the coordination of the PNI, the acquisition of vaccines and the needed materials (syringes and needles) [149]. The states are responsible for the organization of the distribution of vaccines, needles and syringes. Finally, the municipalities are responsible for the execution of the vaccination and the management of the municipal stock of vaccines, as stated in the National Plan of COVID-19 Vaccination Operationalization (*Plano Nacional de Operacionalização da Vacinação contra COVID-19*, PNO) [131].

The MoH publishes a Distribution Schedule (DS) each time one or multiple vaccine companies deliver vaccines to the government, indicating the amount of doses corresponding to each state. The distribution of vaccine doses is organized by the MoH following technical criteria, taking into account the number of doses made available by the vaccine laboratories. The vaccines are delivered to each state proportionally to the number of inhabitants, and are administered at the municipality level. As two doses are needed to complete the vaccination scheme for some vaccines, for example, the Sinovac/Butantan vaccine, and the time between doses is from two to four weeks, the second dose is meant to be stored at the state level. The flow of vaccines between the state and the municipality needs to be mutually agreed.

A 5% operational loss was considered in each DS. Additionally, from the second DS, 5% of the received doses were attributed to elderly people of the Amazonas State due to the high mortality count in this region at this date. From the fourth DS, this 5% was called the "strategic found", and it was maintained in the following DS. Taking into

consideration that the interval between dose one (D1) and dose two (D2) is bigger in the AstraZeneca vaccine than in the Sinovac vaccine (8 to 12 weeks vs. 4 weeks), the distribution of AstraZeneca vaccine started taking place at two times. However, due to the deterioration of the epidemiological situation, a change was made from the ninth DS, distributing only D1 of the Sinovac vaccine to some priority groups in order to reduce the risk of infection. The MoH is the one in charge of taking this into account in the organization of the following delivery of second doses before the limit of four weeks [131].

### 3.3.5. Prioritization of Target Groups

The most recent version of the PNO was released by the MoH in 15 July 2021 (edition 9). The vaccination strategy with the priority groups of the National Vaccination Campaign against COVID-19 published in the aforementioned plan was developed by the PNI with the support of the Technical Advisory Board on Immunization and Communicable Diseases (*Câmara Técnica Assessora em Imunização e Doenças Transmissíveis*) and the WHO Strategic Advisor Group of Experts on Immunization [150,151]. In this plan, the MoH identifies social vulnerability as a risk factor for both suffering more severe symptoms of the disease and dying. As previously mentioned, in this category of "groups with higher social vulnerability" are included: Afro-descendants and Indigenous living in Indigenous Lands, mostly due to the high level of cohabitation and the already existing high level of comorbidities, not only chronic but also infectious. Not to forget the constraints in terms of geographical access to healthcare services, with patients enduring a full day of travel to reach their nearest healthcare service. Other groups considered in the vaccination strategy with higher risk of exposure to the virus and that face social and economic vulnerabilities are: people living on the streets, refugees living in shelters, people with permanent disabilities and populations deprived of freedom [131].

The main objectives of the Brazilian National Vaccination Campaign are to preserve the continuity of the healthcare facilities, to protect the population at higher risk of severe symptoms, to protect the vulnerable population that might suffer the greatest impact and preserve the functioning of the essential services. According to the PNO, in case new scientific evidence about the virus, the vaccines or the epidemiological situation are found, the priority groups are susceptible to change, and new priority groups or subdivision of already existing priority groups might be considered. The General Coordination of the PNI published the technical notes 6/2021 and 17/2021 with the initial prioritization of target groups [80,152]. Moreover, the MoH releases Distribution Schedule reports with the distribution patterns, where they specify the priority groups that should be addressed in each vaccination schedule [153]. Due to insufficient available doses to vaccinate the whole of the population in these groups simultaneously, the PNI released the technical note 155/2021, as a complementary piece to the previous notes, defining the population included in the "priority of the priority" groups and defining the order that should be followed [153]. The estimations of the population included in each category and the specific definition of each category are specified in the technical note. For the groups that are defined taking comorbidities into consideration, the types of comorbidities are specified in Table 1 of the Vaccination Plan. Additionally, for the definition of some priority groups, the *Benefício de Prestação Continuada* (BPC)—continuous cash benefit— is used as a parameter.

The order of the priority groups, with the estimation of the population included in each group is: (1) institutionalized people aged 60 years or older (156,878); (2) institutionalized people living with disabilities (6472); (3) Indigenous Peoples living in Indigenous lands (413,739); (4) health workers (6,688,197); (5) people aged 90 years or older (893,873); (6) people from 85 to 89 years old (1,299,948); (7) people from 80 to 84 years old (2,247,225); (8) people from 75 to 79 years old (3,614,384); (9) Traditional *Ribeirinhas* peoples and communities (286,833); (10) Traditional *Quilombolas* peoples and communities (1,133,106); (11) people from 70 to 74 years old (5,408,657); (12) people from 65 to 69 years old (7,349,241); (13) people from 60 to 64 years old (9,383,724); (14.1) people with comorbidities from 18 to 59 years old (18,218,730); (14.2) people with permanent disabilities with BPC from 18

to 59 years old (1,467,477); (14.3) pregnant and postpartum women from 18 to 59 years old (2,488,052); (15) permanently disabled people from 18 to 59 years old without BPC (6,281,581); (16) homeless people from 18 to 59 years old (140,559); (17) employees of the prison system except healthcare workers (108,949) and populations deprived of freedom (753,966); (18) basic education workers (2,707,200); (19) higher education workers (719, 818); (20) security and rescue forces (584,256) and the Armed Forces (364,036); (21) collective road passenger transport workers (678,264); (22) subway and railway transport workers (73,504); (23) air transport workers (116,529); (24) waterway transport workers (41,515); (25) truck drivers (1,241,061); (26) port workers (111,397); (27) industrial workers (5,323,291); (28) urban cleaning and solid waste management workers (227,567).

In case a municipality reaches all the population of the priority groups recommended by the actual Vaccination Distribution Schedules and the PNO, the vaccination may be moved forward to the next prioritized group indicated in the PNO [154].

### 3.3.6. Current Vaccination Coverage

As of 18 July 2021, 117,787,993 doses had been applied throughout the country, of which 55,000,246 doses are from Oxford–AstraZeneca, 45,943,020 doses from Sinovac/Butantan, 12,871,027 doses from Pfizer/BioNTech and 3,973,700 doses from Janssen (Johnson & Johnson). The charts indicate the proportion of each one of them (Figures S4 and S5).

Additionally, 42.5% of Brazilian population are partially immunized since they received a first dose (Figure S6), while 16.2% can be considered full immunized since they received either one single dose of Janssen vaccine or two doses of the others. Hence, 41.3% of Brazilian population had received no doses at the time of writing.

## 4. Discussion

This integrative review is a snapshot of the development of the Brazilian vaccination campaign and provides insights into its progress and limitations. Unlike some countries that experienced multiple waves, Brazil only experienced two officially recognized waves, because there was never a significant reduction in cases and deaths, and it could be argued that the second wave is only now starting to diminish. As the country dealt with deadly COVID-19 outbreaks, Brazil faced initial challenges in the procurement of different COVID-19 vaccines and in the development of a national vaccination campaign [76]. The lack of coordination at the federal level, the absence of federal government support for scientific research and the lack of endorsement and commitment to the mitigation of the COVID-19 pandemic set the country off to a challenging start [81]. However, Brazil has a strong universal healthcare system, a strong immunization program and strong democratic institutions. Accordingly, the country began its vaccination campaign on January 18th 2021, has signed contracts with six vaccine suppliers so far and is scaling up the domestic production of vaccines [38,130,134]. Brazil is therefore catching up and is utilizing its world-renowned immunization program. Indeed, at the time of writing, Brazil had administered the largest number of doses in Latin America (over 93 million) [155].

However, challenges remain regarding the rollout and implementation of the vaccination plan. First, Brazil is a country where inequalities are present at every level of society and health is no exception. The COVID-19 pandemic has further widened those disparities. Will the vaccination campaign be able to compensate, at least partly, for them? The Ministry of Health has included "social vulnerability" as a risk-group in its vaccination strategy. This allows groups such as *Quilombolas* Communities and Indigenous living in Indigenous Lands to have priority access to the vaccine [38]. The implementation of this measure should be closely followed in future research, especially since these populations are often more difficult to reach logistically. Second, despite Brazil's strong institutions, the overall management of this pandemic has also highlighted the fragility of democracies and the barriers that one political figure can create to the application of evidence-based measures. It seems as though Jair Bolsonaro has now changed his discourse and has started supporting the vaccination campaign. However, the scientific community needs

to continue to investigate the country's vaccination strategy and implementation to make sure the maximum effort is done for the health of the Brazilian population.

We have identified certain limitations to this case-study integrative review. First, the COVID-19 epidemiological situation and its management, including the vaccination, is a dynamic process, which is constantly evolving. This paper is a snapshot of the situation at the time of writing (July 2021) but many of these data are likely to change in the future. It is necessary to note that the data which were used for this paper also sometimes lack transparency and this case-study should therefore be seen as a general overview and not a definite picture. In addition, some inconsistencies have been identified in the distribution schedules of vaccines published by the Brazilian government that may lead to inaccurate data in terms of dates and number of distributed doses.

The COVID-19 vaccination campaign in Brazil has faced many challenges but has highlighted the strong institutional mechanisms ensuring the procurement and distribution of the vaccines. Future research should investigate how the vaccination strategy evolves and most importantly how it is implemented, especially from an equality standpoint.

**Supplementary Materials:** The following are available online at https://www.mdpi.com/article/10.3390/epidemiologia2030026/s1, Figure S1: Seven-day average of daily new confirmed COVID-19 cases, Figure S2. Seven-day average of daily new confirmed COVID-19 cases "Coronavirus Pandemic (COVID-19)", Figure S3: COVID-19 mortality rate per 100,000 inhabitants (A), incidence rate per 100,000 inhabitants (B), lethality rate per 100,000 inhabitants (C) and overall lethality (D) in each Region of Brazil, Figure S4: Applied doses according to producer as of 19 July 2021, Figure S5: Rate of population who has received one dose (green, partially immunized), two doses (blue, immunized) or one dose of Janssen (orange, immunized) as of 19 of July 2021, Figure S6: Percentage of the population who received the first Covid-19 vaccine shot as of 18 July 2021.

**Author Contributions:** Conceptualization, L.B.-S., A.N.-H. and L.S.; Methodology, L.B.-S., A.N.-H. and L.S.; Validation, N.S.G., R.A.T.d.A. and M.S.M.; Writing—original draft, L.B.-S., A.N.-H. and L.S.; Writing—review and editing, L.B.-S., A.N.-H., L.S., N.S.G., R.A.T.d.A. and M.S.M. All authors have read and agreed to the published version of the manuscript.

**Funding:** This research received no external funding. N.S.G. received a postdoctoral scholarship from the National Institute of Science and Technology for Health Technology Assessment (Instituto de Avaliação de Tecnologias em Saúde—IATS)/National Council for Scientific and Technological Development (*Conselho Nacional de Desenvolvimento Científico e Tecnológico*—CNPq) (grant number 465518/2014-1). M.S.M. is a member of IATS/CNPq.

**Institutional Review Board Statement:** Not applicable.

**Informed Consent Statement:** Not applicable.

**Data Availability Statement:** This is a case study based on an integrative review. No new data were created or analyzed in this study. Data sharing is not applicable to this article.

**Acknowledgments:** The authors would like to express their gratitude to Antoine Flahault and Liudmila Rozanova for their support, as well as to Pablo Perel for his valuable contribution to this manuscript.

**Conflicts of Interest:** The authors declare no conflict of interest. The funders had no role in the design of the study; in the collection, analyses, or interpretation of data; in the writing of the manuscript, or in the decision to publish the results.

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
