# Peer review of "The COVID-19 Vaccination Strategy in Brazil—A Case Study"

_epidemiologia, doi:10.3390/epidemiologia2030026_

Round 1
Reviewer 1 Report
The article is very extensive. An article can be extensive if the contents is all significant towards the proposed objective. The contents is interesting and improves the knowledge of the reader about geography and demography of Brasil. However, it is less appropriate for a scientific article. If the reader don't know about Brasil hardly will take notice of the reality in Brasil regarding the pandemic on the other side if wants to analyse the pandemic is because already knows at least part of the reality of Brasil. In my view the reader will get a lot of information that will not be useful for the analysis, as in fact, there is no correlation established between the information presented in the first part of the document and its impact on the pandemic and response by the health services. Section 3 is improperly named results. Those are not results, are a series of facts or informations that the authors consider relevant to mention. As said, there is much information that can be interesting for those willing to know about Brasil but as for the article it is not established a correlation about those facts (and political considerations) and the pandemic. Also, the conclusions seem to contradict the reporting of several weaknesses reported along the document. As for the contents, it is valuable in reporting about covid-19 situation and the regarding the vaccination however it needs to have some work to become more consistent and less divergent in its contents.
Author Response
Dear editor,
We appreciate the reviewers’ comments and suggestions, as their contribution greatly improved the quality of our manuscript. Please find following our response to the comments to the article entitled “The Covid-19 Vaccination Strategy in Brazil — A Case Study”.
First, we would like to specify that we are preparing a special issue of the Journal which will be a series of case studies reviewing the literature on vaccines in a set of selected countries. This paper from Brazil is one of it, and should not be assessed as a research article for regular issues of the Journal but specifically dedicated to the special issue on COVID vaccines.
Regarding the first comment about the relevance of the first part, we included the information on Brazil to provide a background, so the reader can understand the whole case study. We consider important to highlight the extensive territory, the large population, the economic situation, all the inequalities and the difference among Brazilian regions as well as politics and corruption, as all these factors impact on the barriers for a well developed COVID-19 vaccination program, in a country where the national vaccination program was previously well known to be effective. We agree with the reviewer and summarized it, as requested.
The result section was renamed as “Case Study”.
Regarding the comment about the correlation between the political considerations and the pandemic, we consider that the political considerations are key to understand the barriers for a faster and more effective COVID-19 vaccination program. We consider very important to keep them. We summarized the information, as requested.
The conclusions were modified, as suggested.
As for the last comment, the text was revised, as suggested.
Thank you for the feedback.
Kind regards,
Llanos Bernardeau-Serra, Agathe Nguyen-Huynh, Lara Sponagel, Nathalia Sernizon Guimarães, Raphael Augusto Teixeira de Aguiar and Milena Soriano Marcolino
Reviewer 2 Report
Bernardeau-Serra And colleagues summarize the begin of the Covid-19 in Brazil and following the vaccine strategy and campaign in Brazil. Interestingly, Brazil was one of the countries who started rapidly the vaccination of Sinovac and Moderna and as the largest country in latin america, Brazil belongs de facto to the countries (beside USA) who succeeds with the vaccination strategy. For such a success, there must be a clear and straightforward program to reach each person. Thus, the authors have presented the facts from the literature how the Brazilian politics followed such a strategy. Overall, the presentation of the content of the manuscript is good but there are several issues which has to be addressed:
1- The article does not consist any new results. It summarizes the literature. It does not fit the criteria for a research article, maybe a review. Please discuss this issue with the journal.
2- the whole manuscript is too long and redundant, particularly the case presentation part. Please short.
3- the abstract section is too short. Please provide in detail what the material and methods and the main results are.
4- a map of Brazil reflecting regions with the incidence rates of Covid-19 and in counterpart the vaccination would be good
5- please describe the first infected and first vaccinated person.
6- Ll. 24-28 must be cited
7-please discuss why Sinovac firstly used and not moderna/biontech pfizer. In fact, the protection rate of Sinovac is currently and still low.
8- please discuss in detail the covid-19 infection phased in brazil. How many ‚waves‘ were observed and when it was stopped after vaccination.
Author Response
Dear editor,
We appreciate the reviewers’ comments and suggestions, as their contribution greatly improved the quality of our manuscript. Please find following our response to the comments to the article entitled “The Covid-19 Vaccination Strategy in Brazil — A Case Study”.
First, we would like to specify that we are preparing a special issue of the Journal which will be a series of case studies reviewing the literature on vaccines in a set of selected countries. This paper from Brazil is one of it, and should not be assessed as a research article for regular issues of the Journal but specifically dedicated to the special issue on COVID vaccines.
Regarding the second and third comment, the text is now summarized and the abstract modified, as suggested. Moreover, maps were include.
Regarding comment 5, first infected and first vaccinated person were cited. There are now on line 894 and line 2257, respectively. Ll. 24-28 were cited.
Regarding comment 7, the official explanation in Brazil is that some Brazilian politicians preferred and made a Brazilian solution, and the vaccine manufacture laboratories in Brazil are equipped to produce vaccines such as Sinovac which are traditional vaccines. The infrastructure and technology is not ready to produce RNAm vaccines. Furthermore, the federal government initially had no interesting in signing the contract with Pfizer, as described in the manuscript. There are ongoing investigations into how the government handle vaccine acquisition.
Regarding comment 8, unlike some countries that experienced multiple waves, Brazil only experienced two officially recognized waves, because there was never a significant reduction in cases and deaths and it could be argued that the second wave is only now starting to diminish.
Thank you for the feedback.
Kind regards,
Llanos Bernardeau-Serra, Agathe Nguyen-Huynh, Lara Sponagel, Nathalia Sernizon Guimarães, Raphael Augusto Teixeira de Aguiar and Milena Soriano Marcolino
Round 2
Reviewer 1 Report
Some comments I even agree. My comments are in the direction of a typical scientific article. If you say it is a special issue and must have that socio-demographic information, then fine.
Reviewer 2 Report
Dear authors,
Thanks for the detailed revision of the manuscript. The quality significantly increased and all my concerns were well adressed. Good job!